# A Machine Learning Model for the Prognosis of Pulseless Electrical Activity during Out-of-Hospital Cardiac Arrest

**DOI:** 10.3390/e23070847

**Published:** 2021-06-30

**Authors:** Jon Urteaga, Elisabete Aramendi, Andoni Elola, Unai Irusta, Ahamed Idris

**Affiliations:** 1Department of Communications Engineering, University of the Basque Country, 48013 Bilbao, Spain; elisabete.aramendi@ehu.eus (E.A.); unai.irusta@ehu.eus (U.I.); 2Biocruces Bizkaia Health Research Institute, Cruces University Hospital, 48903 Baracaldo, Spain; 3Department of Mathematics, University of the Basque Country, 48013 Bilbao, Spain; andoni.elola@ehu.eus; 4Department of Emergency Medicine, University of Texas Southwestern Medical Center, Dallas, TX 75390, USA; ahamed.idris@utsouthwestern.edu

**Keywords:** out-of-hospital cardiac arrest (OHCA), electrocardiogram (ECG), thoracic impedance (TI), pulseless electrical activity (PEA), return of spontaneous circulation (ROSC)

## Abstract

Pulseless electrical activity (PEA) is characterized by the disassociation of the mechanical and electrical activity of the heart and appears as the initial rhythm in 20–30% of out-of-hospital cardiac arrest (OHCA) cases. Predicting whether a patient in PEA will convert to return of spontaneous circulation (ROSC) is important because different therapeutic strategies are needed depending on the type of PEA. The aim of this study was to develop a machine learning model to differentiate PEA with unfavorable (unPEA) and favorable (faPEA) evolution to ROSC. An OHCA dataset of 1921 5s PEA signal segments from defibrillator files was used, 703 faPEA segments from 107 patients with ROSC and 1218 unPEA segments from 153 patients with no ROSC. The solution consisted of a signal-processing stage of the ECG and the thoracic impedance (TI) and the extraction of the TI circulation component (ICC), which is associated with ventricular wall movement. Then, a set of 17 features was obtained from the ECG and ICC signals, and a random forest classifier was used to differentiate faPEA from unPEA. All models were trained and tested using patientwise and stratified 10-fold cross-validation partitions. The best model showed a median (interquartile range) area under the curve (AUC) of 85.7(9.8)% and a balance accuracy of 78.8(9.8)%, improving the previously available solutions at more than four points in the AUC and three points in balanced accuracy. It was demonstrated that the evolution of PEA can be predicted using the ECG and TI signals, opening the possibility of targeted PEA treatment in OHCA.

## 1. Introduction

Out-of-hospital cardiac arrest (OHCA) is a major public health problem, with an estimated incidence between 350,000 and 700,000 cases per year in Europe and survival rates below 10% [1,2]. A patient in cardiac arrest abruptly looses respiratory and cardiovascular functions and, if untreated, dies within minutes. An early recognition of OHCA and prompt treatment are therefore key for survival. In the prehospital setting, bystander cardiopulmonary resuscitation (CPR) contributes to maintaining artificial blood flow through ventilation and chest compressions until more advanced therapy is available. For instance, when the presenting heart rhythm is ventricular fibrillation (VF), an electrical defibrillation shock within the first five minutes from OHCA onset raises survival rates by 50–70% [2,3].

The best course of treatment for OHCA depends on the heart rhythm of the patient, which can be determined using an electrocardiogram (ECG) [4]. In the preshopital setting, the heart function is monitored by the emergency medical system (EMS) personnel using monitor defibrillators. Unfortunately, by the time the EMS personnel arrives on scene, VF is the presenting rhythm in only 11–37% of OHCA cases [5,6]. A frequently presenting rhythm is pulseless electrical activity (PEA), with recorded incidences of 20–30% out of hospital [7,8,9] and up to 40–60% in hospital [10,11], as well as much lower survival rates [7,12,13,14,15]. PEA is characterized by the dissociation of the electrical and mechanical activities of the heart. Therefore, a patient in PEA presents apparent heartbeats in the ECG with discernible QRS complexes, but without effective ventricular wall movement. Thus, there is no palpable pulse and an insufficient blood flow [7]. EMS personnel provide CPR and pharmacological treatment to revert PEA and achieve return of spontaneous circulation (ROSC), but treatment depends on the characteristics of PEA. Consequently, directions for understanding the mechanism and stratification of PEA have been addressed by clinical consortia and efforts to predict, prevent, and manage PEA encouraged [7,13,15,16].

PEA states can grossly be classified into pseudo-PEA or true-PEA [17]. In pseudo-PEA, the electrical activity of the heart produces a small mechanical activity, albeit insufficient for a palpable pulse. In true PEA, there is no mechanical cardiac activity [16,18]. The two stages of PEA have different prognoses and treatments [7,18,19,20], and their distinction is of great clinical interest to predict the hemodynamic evolution of PEA, as well as whether the patient will recover ROSC.

Several contributions have proposed the use of ECG features to differentiate PEA with favorable evolution to ROSC (faPEA) from PEA with unfavorable evolution to ROSC (unPEA). The heart rate (HR) and the width of the QRS complex during PEA have been extensively investigated in both in- and out-of-hospital cardiac arrest, but with contradictory conclusions [12,13,14,15]. In these studies, ECG data were manually annotated, and no automatic method has been proposed yet to discriminate faPEA from unPEA. Additionally, the thoracic impedance (TI) measured through the defibrillation pads reflects changes in tissue density and fluid content in the thoracic region and thus presents a small, but discernible component associated with blood flow [21]. TI has been successfully used to discriminate PEA from rhythms associated with ROSC, by extracting the impedance circulation component (ICC), which reflects blood flow during ROSC [22,23]. In fact, models combining ECG and TI have been proposed to predict immediate rhythm transitions during OHCA [24] and to discriminate rhythms in OHCA [25], and in a preliminary study, a model combining an ECG and a TI feature showed promising results for the discrimination of faPEA and unPEA on a limited dataset [26].

This study introduced a new model to discriminate faPEA from unPEA based on comprehensive automatic feature extraction from the ECG and TI signals using various signal analysis domains. An advanced random forest (RF) classifier was then used to efficiently combine those features and improve the accuracy of the diagnosis. A comprehensive dataset of OHCA episodes was used for the analysis. The results showed that a combination of ECG and TI features substantially improved the accuracy of the models, which could be used to assist EMS personnel in evaluating the hemodynamic state of the patient and deciding the optimum resuscitation treatment.

## 2. Data Collection

The dataset used in this study was a subset of a larger dataset of OHCA episodes recorded by the Dallas-Fortworth Center for Resuscitation Research (Dallas, TX, USA). Every episode had concurrent ECG (250 Hz, resolution = 1.03 mV) and TI signals (200 Hz, resolution = 0.74 mΩ) recorded by the defibrillation pads of a HeartStart MRx defibrillator (Philips Healthcare, Andover, MA, USA).

The dataset consisted of 260 episodes of patients in PEA, of which 107 recovered ROSC and 153 did not. ROSC recovery was certified by clinicians on site and further revised by visual inspection of the episodes. Cases ending in ROSC had confirmed long periods without CPR after recovery of pulse, while cases without ROSC had CPR until the end of the episode. PEA onset was identified in the episodes as the first occurrence of an organized rhythm (QRS complexes) during CPR. PEA segments of 5 s in duration, separated by at least 1 s, and including the ECG and the TI were identified during the first 10 min after PEA onset. Segments were extracted in the pauses of chest compressions, identified in the TI [27,28], with no artifacts due to compressions in the signals. Figure 1 shows an example of an episode in which PEA evolved to ROSC (in green). Chest compression activity is visible in the TI signal, and PEA segments (in blue) were only selected during the intervals without chest compressions to avoid artifacts in the ECG.

A total of 1921 PEAs were extracted, a median (interquartile range, IQR) of 4 (6.5) segments per episode. The segments in the ROSC episodes were labeled as faPEA and those in the non-ROSC episodes as unPEA. There were a total of 703 faPEA segments, 4 (5.8) per episode; and 1218 unPEA segments, 5 (7) per episode. Figure 2 shows examples of the faPEA and unPEA segments. As shown in the figure, the faPEA segment presents a more regular ECG with narrower QRS complexes of larger amplitude and a higher heart rate. Moreover, it also presents TI components and an ICC waveform correlated with the heartbeats.

## 3. Methods

The algorithm to discriminate faPEA from unPEA consisted of the three stages shown in Figure 3. The first stage was an ECG and TI signal-processing stage, where the ECG and TI signals were resampled to a common sampling rate of $f_{s}=250\;\mathrm{Hz}$ and then denoised to obtain ${\hat{s}}_{\mathrm{ECG}}\left(n\right)$ and ${\hat{s}}_{\mathrm{TI}}\left(n\right)$. The impedance ICC component, $s_{\mathrm{ICC}}\left(n\right)$, was then extracted from ${\hat{s}}_{\mathrm{TI}}\left(n\right)$ by applying adaptive filtering and denoised to obtain ${\hat{s}}_{\mathrm{ICC}}\left(n\right)$. In the second stage, a set of waveform features was computed from the denoised ECG and ICC signals. Finally, in the third stage, these features were fed to an RF classifier to discriminate faPEA from unPEA segments.

### 3.1. Processing of ECG and TI Signals

#### 3.1.1. ECG Processing

The ECG signal was denoised using the stationary wavelet transform (SWT) as proposed by Isasi et al. for OHCA rhythms [29,30]. An 8-level SWT decomposition was used with a Daubechies-4 mother wavelet and soft thresholding. Detail coefficients $d_{3}$ to $d_{8}$ were used to reconstruct the denoised ECG, which corresponds to an analysis band of 0.5–31.25Hz, a typical band for ECG analysis in OHCA [23,29].

#### 3.1.2. TI Processing and ICC Extraction

The TI measured through the defibrillation pads may show different components due to: baseline wandering, chest compressions and ventilation during CPR, the circulation component in the pulsed rhythm, additional noise/artifacts due to movement, electrode–skin contact, etc. [31]. The segments of the database were extracted during pauses of chest compressions, so the TI signal was bandpass filtered (0.8–10Hz) to remove baseline fluctuations, respiration artifacts, and other high-frequency noise [22,32]. Then, the ICC component was extracted, that is the TI component correlated with the ECG heartbeats. Heartbeats were detected in the denoised ECG using the Hamilton–Tompkins algorithm [33], and the instantaneous HR was computed as:(1)$$f\left(n\right)=\frac{1}{f_{s}\left(r_{i+1}-r_{i}\right)}\;\forall n\in \left[r_{i},r_{i+1}\right)$$
where $r_{i}$ is the time instant of the *i*-th QRS complex (R-peak). Using this information, the ICC can be modeled as a Fourier series of *K* harmonics [22,31]:(2)$$s_{\mathrm{ICC}}\left(n\right)=\sum_{k=1}^{K}a_{k}\left(n\right)\cos\left(k\;2\pi f\left(n\right)\;n\right)+b_{k}\left(n\right)\sin\left(k\;2\pi f\left(n\right)\;n\right)$$

The time-varying Fourier coefficients, $a_{k}\left(n\right)$ and $b_{k}\left(n\right)$, were estimated using a Kalman smoother [23]. The Kalman observation and state vectors are then [23,34]:(3)$$\begin{array}{c} x_{n}={\left[a_{1}\left(n\right),\ldots ,a_{k}\left(n\right),b_{1}\left(n\right),\ldots ,b_{k}\left(n\right)\right]}^{T} \end{array}$$(4)$$\begin{array}{c} H_{n}=\left[\cos\left(2\pi f\left(n\right)\;n\right),\ldots ,\cos\left(K\;2\pi f\left(n\right)\;n\right),\sin\left(2\pi f\left(n\right)\;n\right),\ldots ,\sin\left(K\;2\pi f\left(n\right)\;n\right)\right] \end{array}$$

The time-varying Fourier coefficients were assumed to be Gaussian processes with update equations [23,34]:(5)$$a_{k}\left(n\right)=\psi_{n}a_{k}\left(n-1\right)+\omega_{n}$$
(6)$$b_{k}\left(n\right)=\psi_{n}b_{k}\left(n-1\right)+\omega_{n}$$
where $\psi_{n}=\exp\left(-\frac{\lambda }{fs}\right)$ and $\omega_{n}$ is a zero-mean Gaussian process with σ the standard deviation. The update equations are thus:(7)$$x_{n}=\Psi_{n}x_{n-1}+\Omega_{n}$$
where $\Psi_{n}=\psi_{n}\;\cdot \;I_{2K}$, $\Omega_{n}=\sigma \;\cdot \;I_{2K}$ and $I_{2K}$ is the identity matrix of dimension 2K.

The Fourier coefficients (state vector), $a_{k}$ and $b_{k}$, were computed applying the Rauch–Tung–Striebel smoother, with K=5 harmonics, λ=0.05 and σ=0.01, as suggested by Elola et al. [23].

Finally, $s_{\mathrm{ICC}}\left(n\right)$ was denoised using an 8-level SWT (Daubechies-4) with soft thresholding. The $d_{5}–d_{7}$ detail coefficients were used to reconstruct the denoised ${\hat{s}}_{\mathrm{ICC}}\left(n\right)$, which corresponds to the bandwidth 1–8Hz. Figure 2 shows the TI, ICC, and $d_{5}–d_{7}$ detail coefficients for faPEA and unPEA.

### 3.2. Feature Extraction

Since faPEA evolves to ROSC, while unPEA does not, the hypothesis was that faPEA would be more similar to cardiac rhythms with pulse than unPEA. Therefore, faPEA should present more regular interbeat intervals and heart rates, larger ECG amplitudes, wider spectra (narrower QRS complexes), and an ICC with a greater correlation to the heartbeats than unPEA. Therefore, the features used to detect pulse during cardiac arrest were added [23,35,36], as well as the features to quantify signal regularity and spectral dispersion [37,38]. A total of 17 features were computed, 9 from the denoised ECG (${\hat{s}}_{\mathrm{ECG}}$) and 8 from the denoised ICC (${\hat{s}}_{\mathrm{ICC}}$).

#### 3.2.1. ECG Features

The ECG features were (for the detailed calculations, consult [4,29,35,37,38]):The AMSA, the amplitude spectrum area, which is the weighted sum of the amplitudes of the ECG in the spectral domain, and it quantifies the variability and spectral dispersion of the signal. The AMSA was computed as described in [35];${\mathrm{High}}_{\mathrm{power}}$, the power of the ECG in the higher frequency bands; a 17.5–40Hz bandwidth was used [35,38];FuzzEn, fuzzy entropy, which measures the regularity of the signal, computed as described in [35];The SNEO, the smoothed nonlinear energy operator, as described in [37], which measures the local energy content of the ECG;The IQR values of the denoised ECG and its SWT detail coefficients $d_{5}–d_{7}$, which are denoted by $d_{k,\;\mathrm{ECG}}$ for k=5,6,7 [29];${\mathrm{Burg}}_{\mathrm{ECG}}$, the variance of the white noise term of an order-four autoregressive (AR) model estimation of the ECG power spectral density. It measures the goodness-of-fit of the power spectral density to that of spectra concentrated around the fundamental component (HR) and its harmonics [4,39].

#### 3.2.2. ICC Features

The ICC features were (for the detailed calculations, consult [4,22,29,36,37]):${\mathrm{Log}}_{\mathrm{Power}}$, the logarithmic energy (time domain) of the denoised ICC, which has been shown to correlate with ventricular wall movement [22];The SNEO, the smoothed nonlinear energy operator, as described in [37], which measures the local energy content of the ICC;The IQR values of the denoised ICC and its SWT detail coefficients $d_{5}–d_{7}$, which are denoted by $d_{k,\;\mathrm{ICC}}$ for k=5,6,7 [29];${\mathrm{Burg}}_{\mathrm{ICC}}$, the variance of the white noise term of the AR(4) estimation of the power spectral density of the denoised ICC [4,39];${\mathrm{Cross}}_{\mathrm{Power}}$, the cross-power between the denoised ECG and ICC signals, as described in [36].

### 3.3. Building the Classifier

An RF classifier was used, both for feature selection and binary classification of the 5s segments into faPEA/unPEA. RF classifiers have demonstrated good performance and robustness with unbalanced datasets and have the advantage of having an embedded feature ranking/selection through feature importance [40,41].

An RF is an ensemble of *B* decision trees (weak learners), trained using a different bootstrap replica of the original training dataset. The trees are grown using recursive binary splitting, and at each node, $D^{\prime }$ features are randomly selected from the available *D* features for the split. The splitting process is carried out until the tree’s terminal nodes are fed with less than $l_{size}$ observations [40,42]. The final decision of the RF classifier is obtained through a majority vote of those *B* trees.

For this study, an RF classifier with B=500 trees was trained and forced the growth of uncorrelated trees by using a 10% bootstrap replica (with replacement) of the training set for each tree. The number of predictors per node was set to the default $D^{\prime }\;=\;\sqrt{D}$, and the minimum number of observations per terminal node was fixed to $l_{\mathrm{size}}\;=\;5$, as recommended in [23]. To avoid class imbalance, uniform priors were assigned.

For baseline comparisons, other machine learning classifiers were also trained and evaluated. The RF was compared to a logistic regression (LR) classifier and to two support vector machine classifiers with polynomial kernels of second (SVM2) and third order (SVM3). In these models, class imbalance was addressed by weighting the least prevalent class (faPEA) by a factor of 1.5.

### 3.4. Evaluation of the Models

All classifiers were trained and tested using 10-fold cross-validation (CV) with patientwise and stratified data partitions. In this way, training/test data leakage was avoided, and the class imbalance in each fold reflected that of the whole dataset. The CV evaluation of the models was repeated 10 times to statistically characterize the performance of the classifiers.

The classifiers were evaluated using the typical performance metrics for binary classifiers, taking faPEA as the positive class. The following performance metrics were considered: sensitivity (Se), specificity (Sp), balanced accuracy (BAC, the average of Se and Sp), and the area under the receiver operating characteristic curve (AUC).

## 4. Results

Table 1 shows a summary of the statistical distribution of the 17 features for the faPEA and unPEA segments of the complete dataset. The features are ranked by the AUC obtained by using a single-feature LR classifier (evaluated in the 10-fold CV partitions). All features except FuzzyEn showed significant differences for the distributions of the faPEA and unPEA segments (*p* < 0.001, Wilcoxon test), and moderate to good AUC values in the range of 52.9 to 81.6%.

### 4.1. Performance of the RF Classifier

The overall performance of the method is reported in Table 2 in terms of AUC, BAC, and Se/Sp. Two model types were evaluated, those using ECG-only features and those combining ECG and ICC features. For each model, the complete feature set and a reduced optimal feature set based on RF feature importance (see Section 4.2) were used. The models with reduced feature sets showed the best performance, with median (IQR) values of 85.7(9.8)/78.8(9.8)% for AUC/BAC for the ECG+ICC model and 83.2(8.5)/75.7(10.7)% for the ECG-only model. Adding information derived from the impedance (ICC signal) improved the AUC and BAC of the ECG-only models at 2.5 and three points, respectively.

Table 2 also shows the performance of all previous proposals in the literature for the prognosis of the evolution of PEA. All the methods were implemented in MATLAB and then evaluated using this study’s dataset and data partitions. The previous proposals included: (1) a preliminary version of the proposed method based on an RF classifier, but using only one ECG feature (AMSA) and one ICC feature (${\mathrm{Log}}_{\mathrm{Power}}$) [26]; (2) an LR model using ECG-only features proposed by Alonso et al. [24] for the immediate prediction of the evolution of cardiac arrest rhythms, including PEA; (3) single-ECG feature models based on the heart rate [12] and the width of the QRS complexes [14]. In the original studies [12,14], the HR and QRS widths were manually measured, but in an automatic system, these values have to be automatically computed from the ECG. The *wavedec* wavelet-based algorithm was applied both for QRS detection and HR calculations, and for ECG delineation and QRS width calculations [43], and then, we used these features in a single-feature LR classifier. The best solution outperformed all previous proposals by 4–19 points in the AUC and by 3–16 points in BAC. Moreover, the ECG-only solution also outperformed all previous ECG-only solutions by 2–16 points in the AUC and 1.5–14 points in BAC and used a reduced feature set compared to the second-best ECG only model by Alonso et al. [24] (four vs. six).

### 4.2. Feature Selection and Feature Analysis

To analyze how features were ranked, the RF feature importance was used, and the feature selection probability was estimated by adjusting the models of a decreasing number of features ($N_{f}$), from $N_{f}=17,\;\cdots ,\;1$. The selection probability for each feature was measured as the percentage of times it was selected. For each 10-fold CV partition, features were iteratively discarded (in steps of one) by removing the feature with the lowest importance, and the RF models were retrained to rerank the features for the remaining $N_{f}$ features. The process was carried out until a single feature was left. The proportion of times a feature was included for each value of $N_{f}$ is shown in Figure 4.

The most frequently selected features included both ECG and ICC features. The features in the top seven positions were ECG spectral features such as AMSA, Burg, and ${\mathrm{High}}_{\mathrm{Power}}$ and the ICC amplitude/power features such as ${\mathrm{SNEO}}_{\mathrm{ICC}}$, $\mathrm{IQR}(d_{6,\mathrm{ICC}})$, IQR(ICC), and ${\mathrm{Log}}_{\mathrm{Power}}$.

Another important aspect is the performance of the model as a function of $N_{f}$, both to obtain more accurate models by selecting an optimal feature subset, but also to lower the complexity, improve the interpretability, and lower the computational cost of the model. Figure 5 shows the performance of all the classification models (baseline models and the RF classifier) as a function of $N_{f}$, the number of features used in the model. The features included for each $N_{f}$ were those with a higher selection probability (see Figure 4). The best results were obtained for the RF classifier, both in the AUC and BAC, and the RF models showed a stable performance for $N_{f}\geq 6$. As shown in Table 2, the RF classifier with $N_{f}=7$ had the same performance as the RF classifier with the complete set of features.

### 4.3. Time Interval for a Prediction

The time needed from PEA onset to a reliable prognosis is key for a prompt initiation of specific therapies. To analyze the time needed for a prognosis, the faPEA/unPEA classification was performed using only the ECG and TI segments in an interval of $t_{w}\;\left(\min\right)$ from PEA onset, then changing $t_{w}$ from 1 min to 10 min in 1 min steps. Figure 6 shows the AUC and BAC for the different classifiers as a function of $t_{w}$. The RF classifier had the best performance for all time intervals, with AUC and BAC values above 80% and 75%, even for the first minute after PEA onset. As expected, as $t_{w}$ grew as the accuracy of the classifiers improved, since PEA with favorable evolution is closer to conversion to ROSC; however, the improvement in the AUC and BAC was only five points and four points when the interval was extended from 1 min to 10 min; that is, a prompt reliable diagnosis can be obtained, and a specific therapy can be initiated even in the first minute after PEA onset.

Figure 7 shows a combined analysis of the RF performance as a function of $N_{f}$ and $t_{w}$. As shown in the figure, the AUC and BAC increased as the number of features in the model and the analysis interval increased, with AUC values above 85% and BAC values above 78% for $t_{w}\geq 7\;\min$ and $N_{f}\geq 4$.

### 4.4. Analysis of the Classification Errors

The classification errors of the best RF model were analyzed to better understand the limitations and potential future improvements of faPEA/unPEA classification. Figure 8 shows the ECG, TI, and ICC signals for segments with correct classifications and segments with typical patterns leading to classification errors. The top panels show correctly classified segments, despite faPEA having a much lower heart rate than unPEA. In the examples, the TI/ICC signals showed no evidence of mechanical activity for unPEA and activity correlated with the heartbeats in faPEA. The bottom panels show examples of misclassified segments. In the case of faPEA, both the heart rate and the TI/ICC activity were very low, and they corresponded to an episode in which ROSC occurred 38min after PEA onset. In this episode, at the initial stage of PEA, the mechanical activity of the heart was closer to that of unPEA than faPEA. In the case of unPEA, the ECG had a low amplitude and heart rate, as expected for unPEA, but there was noise in the ECG and TI signals in the last part of the segment, which produced a pulse-like ICC signal estimation by the Kalman smoother.

## 5. Discussion

To the best of our knowledge, the proposed method is the first automated method to discriminate PEA rhythms with favorable evolution to ROSC in OHCA data. The algorithm consisted of the extraction of the ICC component of the TI (associated with mechanical wall movement), an ECG and ICC feature extraction phase, and an RF classifier. The solution outperformed previous solutions both in the AUC (four to nineteen points) and BAC (three to sixteen points) [12,14,23,24,26]. Several aspects of the solution explained the better performance. First, the ECG and TI feature set was larger than in previous studies, and the features were carefully selected to reflect or be associated with ventricular wall movement or ROSC. Second, the features obtained from QRS complex segmentation were not used. In nonarrest patients, QRS detection and segmentation are very accurate [43], but their accuracy substantially decreases for cardiac arrest rhythms [35]. For instance, it was observed that the methods based on HR and QRS width presented the lowest performance in part because of the inaccuracies of the automatic algorithms for cardiac arrest data. Third, features obtained from the ICC were added, and these features revealed information on the incipient mechanical activity of the heart in PEA rhythms that converted to ROSC.

The models with reduced the feature sets (seven features for ECG and ICC and four features for ECG-only) had better or comparable performance to those with the complete feature set. Moreover, adding ICC features improved the ECG-only methods by 2.5 points in AUC and 3.1 points in BAC, demonstrating the utility of the TI signal as a surrogate measure of ventricular wall movement [23,26]. A high correlation between the features from the detail components of the ECG and ICC (mean ρ=0.9) and between the spectral features of the ECG (mean ρ=0.7) was observed. An effective feature selection process improves the models, particularly when an exhaustive feature extraction process is carried out [23,29]. More importantly, models with fewer features are computationally less expensive and more explainable. For the RF classifier, using an embedded feature selection based on RF feature importance is an efficient way to obtain close-to-optimal feature subsets.

The time from PEA onset to an accurate prognosis of its evolution is key for the prompt implementation of efficient therapies. In the dataset used in this study, the mean time from PEA onset to outcome (ROSC or no ROSC) was 22 min, and the proposed solution had an AUC and BAC of 81% and 74% within the first minute from PEA onset. Evidently, as time evolved, the accuracy of the prognosis improved, and the AUC and BAC rose to 86% and 79% for an analysis interval of 10 min. In cases in which PEA onset was far from ROSC, errors were more frequent, as shown in Figure 8c for a patient that recovered ROSC 38 min after PEA onset. In any case, there is a clinical tradeoff between the accuracy of the prognosis and the prompt implementation of specific therapies. An alternative approach may be to report the probability of conversion to ROSC as a clinical support tool. Such a probability can be obtained from most machine learning models and in particular in the RF model by computing the proportion of trees with positive faPEA classification [35,41].

The solutions proposed in this study were based on the ECG only and on combined features from ECG and TI (the ICC was derived from TI). In both cases, reasonable tradeoffs between time-to-prognosis and accuracy can be reached. The reason for using these signals is that they are universally available in defibrillators and monitor defibrillators, the equipment used by EMS crews to monitor OHCA patients. All these devices have an ECG channel through the defibrillation pads [35], but not all include a TI signal with sufficient resolution to implement these algorithms [22,23]. Since the proposed algorithms are fully automatic, this means they could be integrated into this equipment as decision support tools for the management of OHCA patients in PEA; that is, they would contribute to a personalized resuscitation treatment, as proposed in the latest resuscitation guidelines [44].

The availability of signals during resuscitation is key to improve the accuracy of automatic algorithms. In particular, the prognosis of ROSC during resuscitation (for all rhythms, not only PEA) is a very active field of research. New and established technologies such as capnography [23,45], cerebral oximetry [46,47], echocardiography [18,48], or point-of-care testing (blood gas analysis) [49] have been explored. A complete up-to-date review is available in [17]. These are, in general, emerging technologies to monitor and guide treatment during OHCA, and only echocardiography and, more recently, capnography have been specifically used to stratify PEA during OHCA [18,23]. In the future, combined algorithms integrating information from all these sources should be explored to improve the prognosis of the evolution of PEA. However, acquiring multimodal OHCA datasets with all these sources of information is complex because OHCA is a critical chronodependent clinical situation treated in a prehospital setting. Therefore, these types of datasets are very scarce and have a limited amount of patients [23].

This study had some limitations. First, the data came from a single type of device, the HeartStart MRx defibrillator. Although the ECGs acquired by different commercial devices have slight differences in bandwidth and resolution, no substantial differences would be expected in the ECG-based model for other devices. Conversely, the TI is acquired by proprietary circuitry, with very different amplitude resolutions and sampling rates. The ICC has a very small amplitude rarely exceeding 100 mΩ, so how well the ICC can be estimated from the TI recorded in other devices needs to be tested. Second, the number of cases included in the study was substantial, but augmenting the dataset’s size would allow the development of more accurate models. In particular, advanced solutions based on deep learning algorithms could also be developed based on features extracted by neural network architectures [50,51,52].

## 6. Conclusions

This study introduced the first machine learning algorithm that discriminates PEA rhythms with favorable evolution to ROSC from those with unfavorable evolution. The proposed algorithm was based on features automatically extracted from the ECG and the TI signal after PEA onset. The RF model proposed outperformed previous solutions, and it demonstrated that both ECG and TI signals contain relevant information for the prognosis of PEA evolution. The results also encourage the development of improved solutions tested on larger datasets. This may lead to decision support tools that assist rescuers in the definition of the best resuscitation treatment during PEA in OHCA, increasing the chances of survival and good neurological outcome. Current commercial defibrillators could benefit from advances in signal processing and machine learning techniques, improving their impact in the course of cardiac arrest resuscitation.

## Figures and Tables

**Figure 1 entropy-23-00847-f001:**
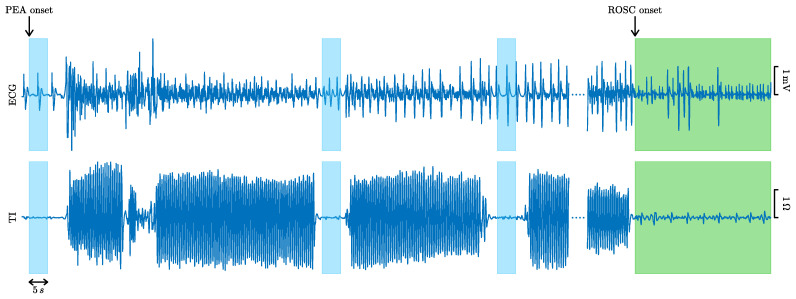
ECG and TI signals of an episode with favorable evolution to ROSC (in green). The 5 s PEA segments extracted from the ECG and the TI are colored in blue.

**Figure 2 entropy-23-00847-f002:**
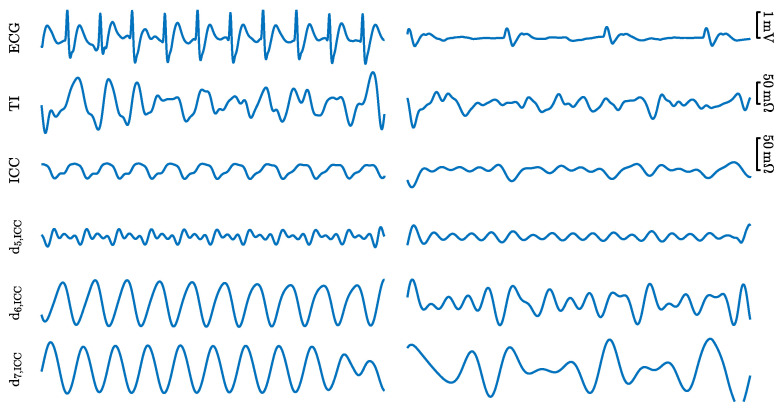
Examples of the signals and components for a 5 s faPEA segment (**left**) and unPEA segment (**right**). From top to bottom: ECG, TI, ICC, and three detail components from the stationary wavelet decomposition of the ICC, $d_{5,\mathrm{ICC}}$, $d_{6,\mathrm{ICC}}$, and $d_{7,\mathrm{ICC}}$.

**Figure 3 entropy-23-00847-f003:**
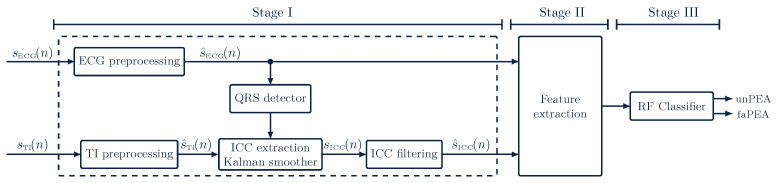
Overview of the faPEA/unPEA classification algorithm. The algorithm consists of three stages: a signal-processing stage, a feature-extraction stage, and a classification stage. The RF classifier uses features from the denoised ECG, ${\hat{s}}_{ECG}\left(n\right)$, and impedance circulation component, ${\hat{s}}_{ICC}\left(n\right)$.

**Figure 4 entropy-23-00847-f004:**
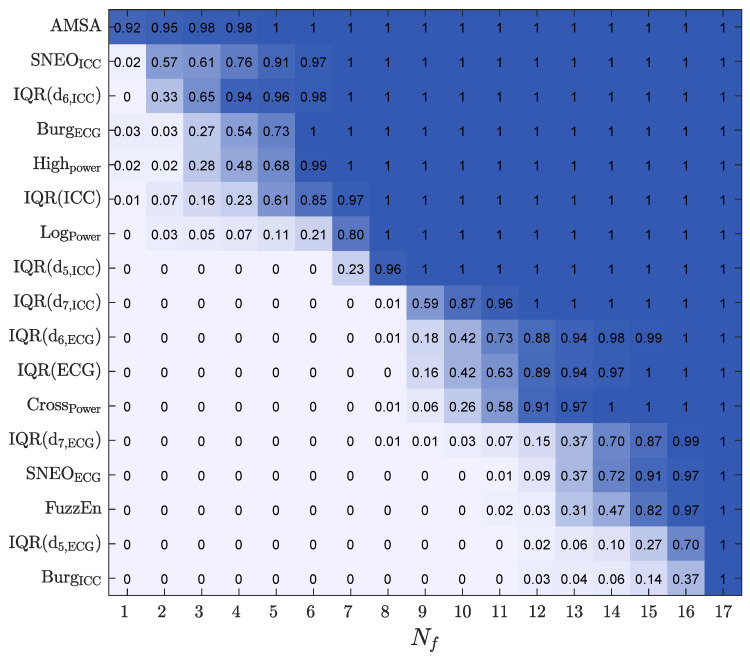
The selection probability for the 17 features, as a function of $N_{f}$, the number of features included in the RF classifier.

**Figure 5 entropy-23-00847-f005:**
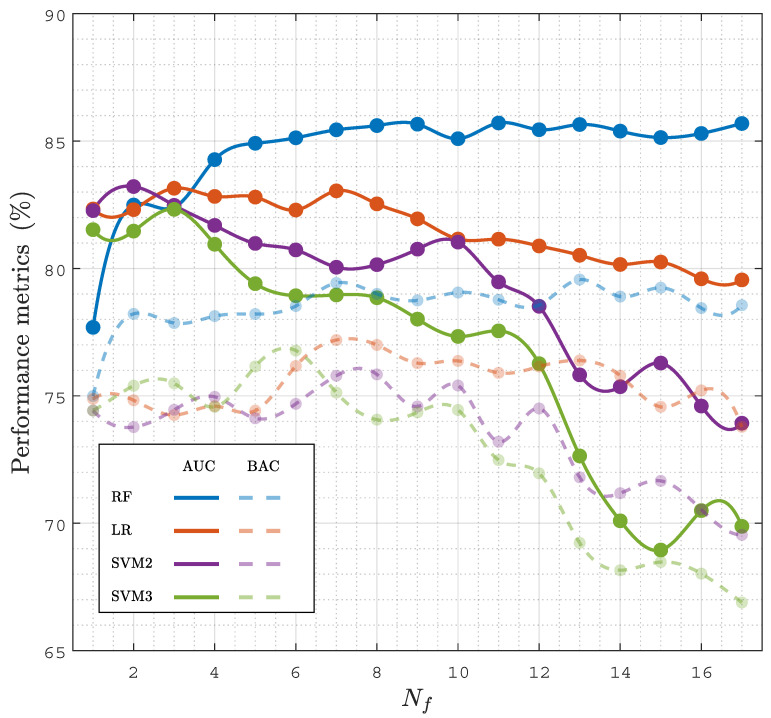
Performance of the classifiers, AUC and BAC, in terms of the number of features, $N_{f}$, considered in the model.

**Figure 6 entropy-23-00847-f006:**
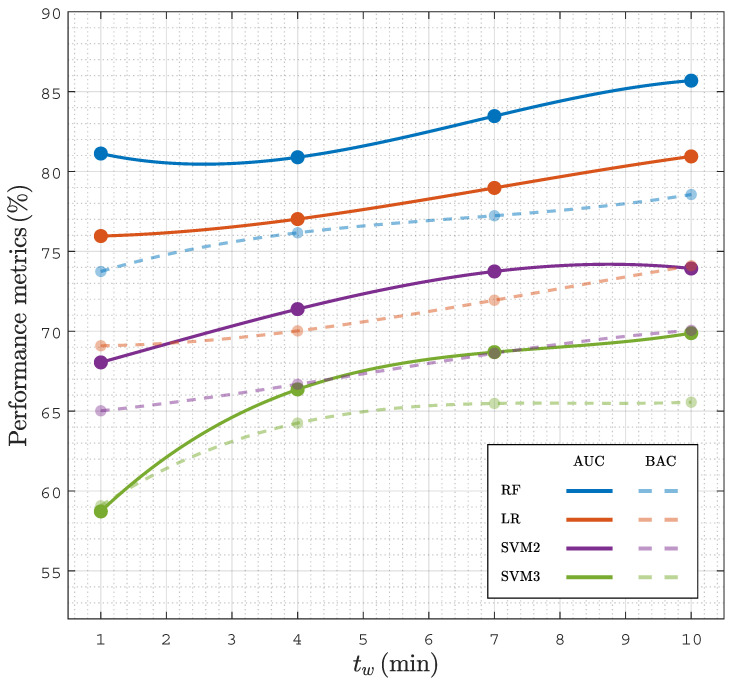
Performance of classifiers in terms of the AUC and BAC as a function of for analysis intervals of $t_{w}$ duration after the onset of PEA.

**Figure 7 entropy-23-00847-f007:**
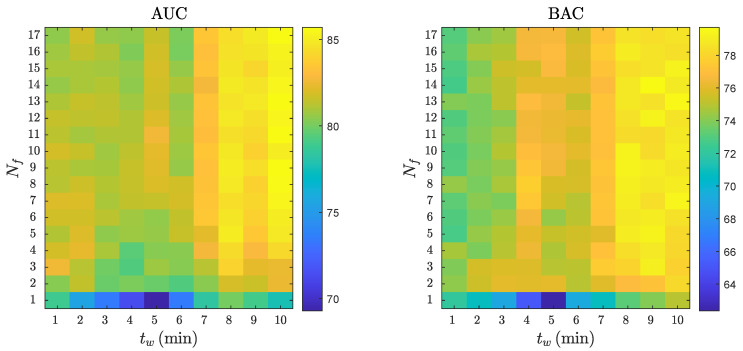
The AUC and BAC of the RF classifier for different analysis intervals, $t_{w}$, and number of features, $N_{f}$.

**Figure 8 entropy-23-00847-f008:**
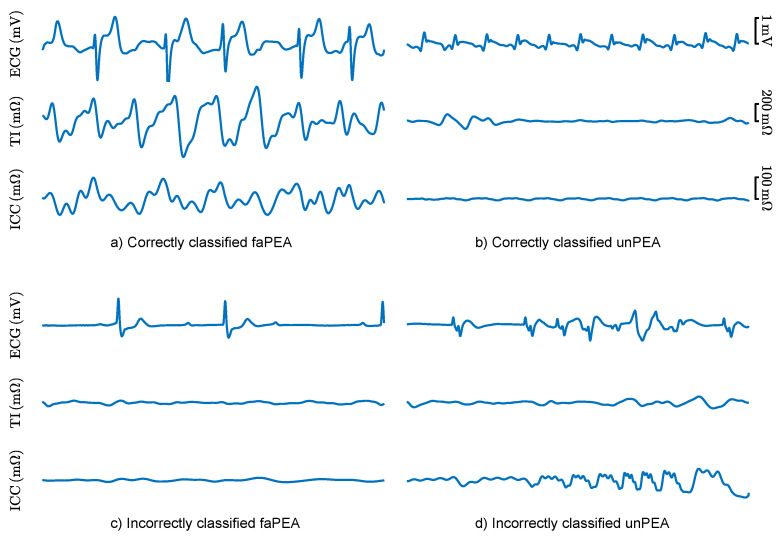
ECG, TI, and ICC signals for 5 s segments of correctly (**top**) and incorrectly (**bottom**) classified faPEA and unPEA segments.

**Table 1 entropy-23-00847-t001:** Median (IQR) values of the features for faPEA and unPEA segments grouped by ECG (left) and ICC (right) features. Features are ranked within each group by the AUC (median, IQR) of a single-feature LR classifier.

ECG Features				ICC Features			
Feature	faPEA	unPEA	AUC (%)	Feature	faPEA	unPEA	AUC (%)
Burg	2.4 ×$10^{-6}$ (3.8 ×$10^{-6}$)	5.6 ×$10^{-7}$ (1.1 ×$10^{-6}$)	81.6 (5.6)	${\mathrm{Cross}}_{\mathrm{Power}}$	1310 (2151)	425 (1083)	71.6 (3.4)
AMSA	31.2 (22.3)	13.1 (14.1)	81.3 (4.9)	IQR($d_{5}$)	22.1 (36.3)	10.5 (30.2)	66.5 (1.6)
${\mathrm{High}}_{\mathrm{Power}}$	74.3 (166.0)	8.3 (24.8)	80.3 (8.1)	SNEO	2930 (10,001)	445 (4427)	65.7 (7.8)
IQR($d_{6}$)	1.1 (1.2)	0.5 (0.6)	72.6 (15.0)	${\mathrm{Log}}_{\mathrm{Power}}$	5131 (2783)	2822 (5259)	64.4 (7.5)
IQR($d_{5}$)	0.31 (0.65)	0.17 (0.29)	71.0 (11.1)	IQR($d_{6}$)	84.2 (136.1)	32.5 (88.9)	64.3 (5.2)
SNEO	0.21 (0.82)	0.06 (0.20)	71.0 (14.4)	IQR	18.6 (26.9)	7.2 (30.5)	61.5 (10.3)
IQR	0.17 (0.17)	0.10 (0.10)	68.8 (14.4)	IQR($d_{7}$)	150.9 (253.8)	66.3 (247.5)	54.9 (13.0)
IQR($d_{7}$)	1.3 (1.5)	1.0 (1.0)	65.2 (12.6)	Burg	0.21 (1.9)	0.05 (0.8)	54.6 (14.4)
FuzzEn	0.22 (0.13)	0.23 (0.14)	52.9 (20.4)				

**Table 2 entropy-23-00847-t002:** Performance of the methods introduced in this study compared to all previous proposals for faPEA/unPEA discrimination. The table shows the median (IQR) values for AUC, BAC, Se, and Sp.

	No. Features	AUC (%)	BAC (%)	Se (%)	Sp (%)
This study(ECG+TI)	17	85.7 (8.6)	77.8 (8.9)	79.8 (11.3)	77.3 (12.1)
This study(ECG)	9	82.1 (9.7)	73.5 (11.2)	79.7 (14.1)	69.0 (15.9)
This study, reduced(ECG+TI)	7	85.7 (9.8)	78.8 (9.8)	80.1 (12.6)	76.7 (13.6)
This study, reduced(ECG)	4	83.2 (8.5)	75.7 (10.7)	78.9 (15.9)	75.7 (11.4)
Urteaga et al. [26]	2	82.0 (10.5)	74.8 (11.3)	77.0 (13.9)	73.5 (14.6)
Alonso et al. [24]	6	81.4 (10.3)	74.4 (8.9)	73.2 (15.1)	77.8 (15.3)
HR [12]	1	67.2 (12.9)	62.1 (11.8)	80.2 (14.5)	45.1 (21.1)
QRS width [14]	1	69.2 (12.9)	67.8 (13.3)	74.8 (20.2)	61.5 (26.6)

## Data Availability

Restrictions apply to the availability of these data. Data was obtained from Dallas-Fortworth Center for Resuscitation Research and are available on request from the corresponding author with the permission of Dr. Idris (Dallas-Fortworth Center for Resuscitation Research).

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
