# Peer review of "A Machine Learning Model for the Prognosis of Pulseless Electrical Activity during Out-of-Hospital Cardiac Arrest"

_entropy, 2021, doi:10.3390/e23070847_

Round 1

Reviewer 1 Report

In this study, the authors developed a random forest-based machine learning model to classify favorable PEA from unfavorable PEA with features extracted from both ECG and ICG signal. The results demonstrate promising accuracy, which is of significance to the clinical prognosis of out-of-hospital cardiac arrest that was along with PEA. The paper is well-written, and almost in the form being accepted for publication.

One minor point: in the second line of section 3.2.1, the “he” typo of “the”?

Author Response

We would like to thank the reviewers for their constructive criticism and thorough revision of our manuscript. Their comments and suggestions have clearly contributed to improve the scientific quality of our manuscript.

Next, the concerns of reviewer 1 and reviewer 2 will be addressed one by one. The text is formatted according to the following criteria:

  1. The reviewer's questions/points are in italics.
  2. The detailed responses to those questions/points are in roman.
  3. The changes in the manuscript and the original version are in italics preceded by the

            line numbers of the revised manuscript.

  1. The changes in the updated manuscript are highlighted in blue.

One minor point: in the second line of section 3.2.1, the “he” type of “the”?

The reviewer is right. It was a mistake we have corrected in line 155.

Reviewer 2 Report

In this work, a novel model to discriminate pulseless electrical activity with favorable evolution from an unfavorable one based on comprehensive automatic feature extraction from the ECG and TI signals using various signal analysis domains is presented. The manuscript is well structured, the methodology exposed seems appropriate and the results presented are really promising. As minor concerns:

  • I do not think it is appropriate to start a section with the expression “This is, […]” at the beginning of the Discussion section.
  • Only chest compressions artifacts are considered in this work. A comprehensive analysis of all interferences that influence the measurements should be included, especially those generated by electromyographic signals.
  • The conclusions are a brief summary of the work carried out. This section should highlight the main findings of the work more broadly.
  • The use of first persons (i.e., ‘we’, ‘their’, possessives, and so on) should be avoided and can preferably be expressed by the passive voice or other ways.

Author Response

We would like to thank the reviewers for their constructive criticism and thorough revision of our manuscript. Their comments and suggestions have clearly contributed to improve the scientific quality of our manuscript.

Next, the concerns of reviewer 1 and reviewer 2 will be addressed one by one. The text is formatted according to the following criteria:

  1. The reviewer's questions/points are in italics.
  2. The detailed responses to those questions/points are in roman.
  3. The changes in the manuscript and the original version are in italics preceded by the

            line numbers of the revised manuscript.

  1. The changes in the updated manuscript are highlighted in blue.

I do not think it is appropriate to start a section with the expression “This is, […]” at the beginning of the Discussion section.

We agree. Now lines 286-287 read:

To the best of our knowledge, the proposed method is the first automated method to discriminate PEA rhythms with favorable evolution to ROSC in OHCA data.

-------------------------------------------------------------------------------------

Only chest compressions artifacts are considered in this work. A comprehensive analysis of all interferences that influence the measurements should be included, especially those generated by electromyographic signals.

We agree with the reviewer that a comprehensive description of the artifacts during CPR was not provided. In the study the segments of the dataset were extracted when chest compressions were not provided, i.e. during pauses of chest compressions. Otherwise the artifacts would severely hide the circulation component of the TI, which is a small component very sensitive to artefacts.

To clarify this, lines 97-98 in section 2 now describe the criterion for segment extraction:

Segments were extracted in the pauses of chest compressions, identified in the TI [27,28], with no artefacts due to compressions in the signals.

and lines 125-130 in section 3.1.2 describe the components of the TI in cardiac arrest episodes. In order to maintain the component correlated with the heart beats, a band-pass filter was applied which removes undesired components:

The TI measured through the defibrillation pads may show different components due to: baseline wandering, chest compressions and ventilations during CPR, circulation component in pulsed rhythm, and additional noise/artifacts due to movement, electrode-skin contact, etc. [31]. The segments of the database were extracted during pauses of chest compressions, so the TI signal was band-pass filtered (0.8-10 Hz) to remove baseline fluctuations, respiration artefacts and other high frequency noise [22, 32].

 ------------------------------------------------------------------------------------

The conclusions are a brief summary of the work carried out. This section should highlight the main findings of the work more broadly.

To answer to this concern, we rewrote the conclusion section emphasizing the achievements of this study. Now lines 352-361 read:

This study introduces the first machine learning algorithm that discriminates PEA rhythms with favorable evolution to ROSC from those with unfavorable evolution. The proposed algorithm is based on features automatically extracted from the ECG and the TI signal after PEA onset. The RF model proposed outperformed previous solutions, and it demonstrates that both ECG and TI signals contain relevant information for the prognosis of PEA evolution. The results also encourage the development of improved solutions tested on larger datasets. This may lead to decision support tools that assist rescuers in the definition of the best resuscitation treatment during PEA in OHCA, increasing the chances of survival and good neurological outcome. Current commercial defibrillators could benefit from advances in signal processing and machine learning techniques improving their impact in the course of cardiac arrest resuscitation.

-------------------------------------------------------------------------------------- 

The use of first persons (i.e., ‘we’, ‘their’, possessives, and so on) should be avoided and can preferably be expressed by the passive voice or other ways.

The English language and style of the manuscript have been thoroughly reviewed avoiding first persons and using the passive voices.
